- Managed aquifer recharge and extraction effects on groundwater level and quality
- dynamics in a typical temperate semi-arid fissured karst system: A multi-method
- quantitative study
- Han Cao<sup>1,2</sup>, Jinlong Qian<sup>1,2</sup>, Huanliang Chen<sup>3,4</sup>, Chunwei Liu<sup>3,4</sup>, Shuai Gao<sup>3,4</sup>, Minghui Lyu<sup>3,4</sup>, Weihong
- Dong<sup>1,2,\*</sup>, Caiping Hu<sup>3,4,\*</sup>

- <sup>1</sup> Key Laboratory of Groundwater Resources and Environments, Ministry of Education, Jilin University,
- Changchun, People's Republic of China
- <sup>2</sup> Institute of Water Resources and Environment, Jilin University, Changchun, People's Republic of China
- <sup>3</sup> 801 Institute of Hydrogeology and Engineering Geology, Shandong Provincial Bureau of Geology &
- Mineral Resources, Jinan, China
- <sup>4</sup> Shandong Engineering Research Center for Environmental Protection and Remediation on
- Groundwater, Jinan, China

Corresponding Author: dongweihong@jlu.edu.cn, caipinghu126@126.com

#### Abstract

Managed Aquifer Recharge (MAR) is an effective approach to mitigate groundwater decline and

spring depletion in karst systems impacted by excessive exploitation. However, the hydrogeological

complexity of karst aquifers makes groundwater quantity and quality highly sensitive to human activities,

posing challenges for MAR implementation. This study develops an integrated multi-method

framework—combining isotopic analysis, flow monitoring, tracer tests, and numerical modeling—to

evaluate the effects of MAR and groundwater extraction on karst aquifer dynamics, with a case study in

the Baotu Spring system (Jinan, China). To enhance the accuracy of recharge rate quantification, an

enhanced isotope mixing model that reduces uncertainties in estimating groundwater recharge ratios from multiple sources was developed, and the MAR rate settings were refined by establishing a quantitative relationship between effective MAR rates and water release rates through river flow monitoring. To improve the solute transport simulations reliability, we conducted field tracer tests to constrain the effective porosity of the karst aquifer - a parameter typically poorly constrained in such systems. Furthermore, we validated the applicability of the equivalent porous media (EPM) model through rigorous hydrodynamic analysis, using field-measured fracture apertures to calculate Reynolds numbers and verify laminar flow conditions. The results demonstrate that surface water contributes >80% of recharge near MAR implementation zones, with MAR efficiency decreasing beyond critical river discharge thresholds. The karst aquifer exhibits laminar flow (effective porosity = 1.08×10<sup>-4</sup>), confirming the validity of the EPM approach. Modeling reveals that MAR significantly raises water tables, though efficiency varies by different MAR sources, and MAR-induced sulfate concentrations must be maintained below 56.5, 197.8, and 339.1 mg/L to meet China's Class I, II, and III groundwater standards, respectively. These findings provide practical guidelines for MAR implementation in temperate semiarid fissured karst systems.

**Keywords:** Managed aquifer recharge (MAR); Temperate semi-arid region; Fissured karst; Multi-41 methods.

#### 1. Introduction

Managed aquifer recharge (MAR) (Sherif et al., 2023), refers to the intentional recharge of aquifers to address the ecological and environmental geological issues caused by excessive groundwater exploitation (Aeschbach-Hertig & Gleeson, 2012; Foley et al., 2011). It has been demonstrated that appropriate recharge can effectively elevate groundwater levels and improve groundwater quality to

some extent (Ajjur & Baalousha, 2021; Alam et al., 2021; Standen et al., 2020).

Karst groundwater constitutes a vital water resource (Hartmann et al., 2014; Medici et al., 2021), with managed aquifer recharge (MAR) in karst systems emerging as a key research focus (Zhang & Wang, 2021). Unlike pore water, karst groundwater is stored in dissolution conduits and fissures, exhibiting high heterogeneity, rapid flow velocities, and concentrated discharge. These properties increase the susceptibility of karst aquifers to anthropogenic impacts on both quantity and quality (Allocca et al., 2014; Lorenzi et al., 2024), complicating MAR implementation. The extreme heterogeneity of karst systems results in spatially variable MAR effectiveness (Daher et al., 2011), with recharge impacts on groundwater levels and quality differing by water source. Rapid flow dynamics (Bakalowicz, 2005) lead to extensive well catchment areas, where over-exploitation can induce large-scale drawdown cones and associated geological risks (Jiang et al., 2019). Furthermore, MAR using contaminated source water may accelerate pollutant transport (H. Cao et al., 2023), jeopardizing groundwater quality (Xanke et al., 2017). Thus, quantitative assessment of MAR and extraction effects on karst groundwater is critical for ensuring sustainable and safe aquifer management. From a global perspective, significant differences exist in karst development and groundwater flow characteristics among different countries and regions. The Baotu Spring karst aquifer in Jinan, China, representing the fissured karst system in the temperate semi-arid region, exhibits remarkable hydrogeological representativeness worldwide (Liang et al., 2018). In these areas, karst aquifers typically develop in Cambrian-Ordovician carbonate formations, with their hydrogeological features being strongly controlled by geological structures. The primary aquifer medium consists of karst fissures formed by well-developed tectonic fissures, ultimately giving rise to a groundwater system dominated by an extensive network of karst fissures (Aliouache & Jourde, 2024; Jiang et al., 2022). In temperate

semi-arid regions, the persistent development of dissolution is constrained by the low permeability of soluble rocks (dominated by fracture flow) and limited hydrothermal conditions, resulting in the prolonged stagnation of underground karst systems at the fracture network stage and hindering their evolution into large-scale cave or conduit systems. Moreover, such regions often feature large karst springs as concentrated discharge points of groundwater (Criss, 2010). Due to seasonal recharge fluctuations (primarily from precipitation) (Bhering et al., 2021), these springs exhibit significant discharge variations. Therefore, scientifically adjusting recharge strategies based on precipitation variability to maintain spring flow constitutes a key research issue.

In China, temperate semi-arid fissured karst groundwater systems similar to the Baotu Spring are predominantly distributed across several northern provinces, including Shandong (Liu et al., 2021), Shanxi (Zhang et al., 2018), Hebei (M. Gao et al., 2023), Henan (Yin et al., 2023) and Shaanxi (Li et al., 2020). Globally, systems exhibiting varying degrees of similarity can be observed in certain regions, notably in the U.K (Agbotui et al., 2020), France (Ballesteros et al., 2020), Germany (Knöll & Scheytt, 2017), Italy (Pagnozzi et al., 2020), the U.S (Criss, 2010) and Canada (Perrin et al., 2011). These regions all face similar challenges related to seasonal drought and karst groundwater pollution. The research on karst groundwater at Baotu Spring and its artificial recharge practices can provide valuable insights for these areas.

Existing research on the effects of managed aquifer recharge (MAR) and extraction on groundwater level and quality has established relatively mature methodologies (Ringleb et al., 2016). However, most approaches remain qualitative or semi-quantitative. Hydrogeochemical and isotopic techniques are widely employed in MAR studies (Akurugu et al., 2022; M. Li et al., 2023). Isotopic tracers are frequently used to identify recharge sources, and the integration of multiple hydrochemical and isotopic

indicators (Guo et al., 2019) allows estimation of source contributions (Deng et al., 2022). Nevertheless, this method faces challenges, including the inherent non-uniqueness of solutions and uncertainty in determining precise isotopic signatures for individual recharge source, which may compromise accuracy. Additionally, the scarcity of long-term isotopic monitoring data restricts the applicability of this approach for analyzing temporal variations in MAR effects.

Numerical simulation serves as an effective method for MAR quantitative analysis (Medici et al., 2021; Ostad-Ali-Askari & Shayannejad, 2021; Zafarmomen et al., 2024). The selection of simulation programs depends on karst aquifer characteristics. While conduit flow process (CFP) models are suitable for well-developed karst systems (Chang et al., 2015), their application is constrained by the requirement for detailed conduit dimension data, particularly in regional-scale modeling (Jourde & Wang, 2023). Previous studies have demonstrated the feasibility of employing a simplified equivalent porous medium (EPM) model without embedded karst conduits for regional groundwater numerical simulations in temperate semi-arid fissured karst systems with limited karst development (Kang et al., 2011; Luo et al., 2020; Scanlon et al., 2003). However, these studies often lack field investigations to verify whether groundwater flow regimes satisfy the laminar flow assumption inherent to EPM models (Agbotui et al., 2020; Medici et al., 2024).

Studies indicate that accurate estimation of effective porosity in karst aquifers is critical when simulating solute transport using the equivalent porous medium (EPM) model (Kidmose et al., 2023; Ren et al., 2018). Overestimation of effective porosity often leads to underestimated groundwater flow velocities, introducing significant errors in pollution control strategies (Medici & West, 2021; Medici et al., 2019). To improve EPM model reliability, effective porosity should be derived from regional-scale hydraulic tests (e.g., tracer test) (Medici & West, 2021; Worthington et al., 2019; Zhu et al., 2020).

Similarly, in MAR studies using numerical simulations, precise determination of groundwater recharge rates is essential for result accuracy (Hartmann et al., 2015). For MAR driven by riverbed infiltration, methods such as infiltration tests (Xi et al., 2015) and riverflow monitoring can quantify recharge rates via hydrodynamics and water-balance principles. Cross-validation of these methods in field studies reduces uncertainty from data limitations, enhancing MAR-related quantitative assessments (Mudarra et al., 2019).

This study proposes an integrated multi-method analytical approach to quantitatively assess the effects of managed aquifer recharge (MAR) and extraction on groundwater levels and quality in temperate semi-arid fractured karst systems. The approach combines coupled numerical modeling of groundwater flow and solute transport with supplementary techniques—isotope analysis, infiltration tests, flow monitoring, and tracer tests—to improve simulation accuracy. Using Jinan's Baotu Spring karst aquifer as a case study, we evaluate how MAR and extraction influence karst groundwater dynamics, aiming to ensure stable regional water levels, long-term water quality security, and sustainable groundwater resource utilization. The specific objectives of this research are as follow: (1) To determine the sources of groundwater recharge and quantify the mixing ratios and spatial distribution of recharge using multi-source data. (2) To quantify the effective infiltration recharge of the MAR segments under varying water release rates for groundwater flow modeling inputs. (3) To estimate the effective porosity of aquifers as a key parameter for groundwater solute transport modeling. (4) To establish a groundwater flow-solute transport model for the study area based on the validated the EPM model's applicability, and to quantitatively evaluate the impacts of MAR and extraction on groundwater level and quality dynamics.

#### 2. Material and Methods

# 2.1. Study area

The study area of this paper is the Baotu Spring area, located in Jinan City, Shandong Province, China, covering about 1,654 km² (Niu et al., 2021) (Fig. 1). The terrain of Baotu Spring area is higher in the south and lower in the north, featuring rolling steep mountains and deep canyons in the south, low mountains and hills in the middle, and Piedmont inclined plains and alluvial plains in the north. The Baotu Spring area is located in the mid-latitude inland area with a warm temperate continental climate. The average annual precipitation from 1951 to 2024 is 690.4 mm, mostly falling between June and September (accounting for 77% of the total).

(a) Geological map

#### (b) Hydrogeological cross-section

Fig. 1 Geological map and hydrogeological cross-section of Baotu Spring area

The main rivers in this region are the Yellow River, Yufu River, Xingji River, and North Dasha River (Fig. 1). The Yellow River forms the study area's northwestern boundary and is mainly used for agricultural irrigation and groundwater recharge. The Yufu River is a seasonal tributary of the Yellow River, and the segments between Zhaiertou and Cuima Villages has excellent permeability, making it an ideal river for MAR (Guo et al., 2019) (Fig. 5(b)). Additionally, the Xingji River in the northeast, though small, also has a permeable riverbed making it suitable for MAR (Fig. 5(b)).

Geologically, the study area is characterized by a northward-dipping monocline predominantly composed of Paleozoic carbonate rock layers. Several large-scale NNW-trending faults are developed within this area. Except for the Dongwu Fault and Mashan Fault forming the eastern and western boundaries of the study area, respectively, the other faults are generally permeable. The stratigraphic units exposed in the study area from south to north, listed from oldest to youngest, are as follows: Archaean Taishan Group metamorphic rocks, Cambrian limestone, Ordovician limestone, and Quaternary loose sediments (Fig. 1).

Exploitable karst groundwater is stored in the Zhangxia Formation of the middle Cambrian, the Chaomidian Formation of the upper Cambrian, and the Majiagou Formation of the Ordovician. In the

northern mountainous and hilly areas, karst groundwater is recharged by precipitation and surface water, flowing northward along strata dips (Zhu et al., 2020). In the northern piedmont alluvial plain, the karst aquifer is buried under Quaternary sediments. Late Mesozoic large gabbro intrusions block northward flow, causing water to rise along fissures and form springs (Niu et al., 2021; Wang et al., 2022), with Baotu Spring being the most popular of them (Guo et al., 2019).

As a crucial water supply source for Jinan, decades of increasing demand for groundwater have led to over-exploitation, causing water level decline and spring drying (S. Gao et al., 2023). To balance water supply and spring protection, Jinan City has implemented the MAR projects using the diverted water from the Yellow River (Kang et al., 2011) and the Yangtze River from the South-to-North Water Transfer Project (Liu et al., 2020) in recent years. Most MAR occurs along segments of the Yufu and Xingji Rivers, with minor MAR through dedicated wells in urban areas (which are no longer used for extraction) (Wang et al., 2017). Notably, excessive flow in the Yufu River may bypass recharge zones, and some diverted water components (e.g., hydrochemical concentrations) may exceed local karst groundwater standards (X. Cao et al., 2023; J. Li et al., 2023; Zheng et al., 2020), posing risks of long-term quality deterioration (J. Li et al., 2023; Zhang & Wang, 2021). Currently, few groundwater exploiting wells remain active, categorized into three groups by location: western suburbs, western Jinan, and eastern suburbs wells. All exploiting wells, MAR wells, and MAR river segments are mapped in Fig. 5(b).

#### 2.2. Groundwater sampling and recharge percentage quantification

In the study area, surface water is the primary source of MAR. To understand the current water quality status of karst groundwater and surface water in the study area, identifying the hydrochemical components in surface water that may influence groundwater quality, estimate the percentage contribution of surface water and precipitation recharge to karst groundwater, and provide a basis for the

groundwater flow and solute transport model setup, sampling and analysis of groundwater and surface water in the study area were conducted in June 2022. The analyzed indicators included total dissolved solids (TDS), sulphate concentration, nitrate concentration, chloride concentration,  $\delta^2$ H‰, and  $\delta^{18}$ O‰. The locations of the sampling points are shown in Fig. 5(b).

After the analysis of groundwater and surface water samples, the origin and recharge sources of karst groundwater were determined by utilizing the  $\delta^2$ H‰ and  $\delta^{18}$ O‰ scatter plot of groundwater and surface water. The analysis of  $^2$ H and  $^{18}$ O requires referencing the Global Meteoric Water Line (GMWL) and the Local Meteoric Water Line (LMWL). The GMWL is given as (Craig, 1961):

$$\delta^2 H = 8\delta^{18} O + 10$$
 (1)

Using the China Meteoric Water Line (CMWL) as the LMWL for the study area, which is:

$$\delta^2 H = 7.7 \delta^{18} O + 7$$
 (2)

Then, to quantitatively analyze the effect of MAR from surface water on karst groundwater,  $\delta^2 H\%$  and  $\delta^{18}O\%$  values were used to determine the proportion of groundwater recharge from surface water. Groundwater in the study area has two main recharge sources: surface water and precipitation (Liu et al., 2021). An improved two-end-member mixing model was employed to calculate the mixing ratios of surface water and precipitation in groundwater samples. Assuming  $\delta^2 H\%$  and  $\delta^{18}O\%$  values for surface water are  $x_s$  and  $y_s$ , for precipitation are  $x_p$  and  $y_p$ , and for groundwater are  $x_g$  and  $y_g$ , the mixing ratios from surface water ( $\eta_s$ ) and precipitation ( $\eta_p$ ) were calculated. The traditional two-end-member mixing model uses either  $\delta^2 H\%$  or  $\delta^{18}O\%$  data to calculate these ratios with the Equation:

$$\begin{cases}
\eta_s = \frac{x_g - x_p}{x_s - x_p} \\
\eta_p = \frac{x_s - x_g}{x_s - x_p}
\end{cases}$$
(3)

Or:

$$\begin{cases}
\eta_s = \frac{y_g - y_p}{y_s - y_p} \\
\eta_p = \frac{y_s - y_g}{y_s - y_p}
\end{cases} \tag{4}$$

Due to the complexity of hydrogeological conditions (there may be unknown recharge sources affecting groundwater isotope values) and the limitations in endmembers selection (isotopic values of precipitation and surface water also vary across different regions), groundwater samples do not completely fall on the mixing line between two end-members in the  $\delta^2$ H‰– $\delta^{18}$ O‰ diagram. For certain samples located far from the mixing line (such as Point A in Fig. 2), calculating the mixing ratio using Eq. 3 or 4 essentially involves projecting sample Point A along the X- or Y-axis to Points A<sub>(3)</sub> or A<sub>(4)</sub>, respectively, which may lead to significantly different results. To address this issue, this study proposes a method for computing the mixing ratio by projecting groundwater sample points onto the two-endmember mixing line in the  $\delta^2$ H‰– $\delta^{18}$ O‰ diagram (it is reasonable to assume that using the closest point on the mixing line, i.e., the orthogonal projection of the sample point A<sub>(5)</sub>, yields a more reliable mixing ratio). The derived Equation for calculating the mixing ratio is as follows:

$$\begin{cases}
\eta_s = \frac{(x_g - x_p)(x_s - x_p) + (y_g - y_p)(y_s - y_p)}{(x_s - x_p)^2 + (y_s - y_p)^2} \\
\eta_p = \frac{(x_s - x_g)(x_s - x_p) + (y_s - y_g)(y_s - y_p)}{(x_s - x_p)^2 + (y_s - y_p)^2}
\end{cases} (5)$$

Fig. 2 Schematic diagram illustrating the method for calculating two-endmember mixing proportions

In this research, the minimum  $\delta^2H\%$  and  $\delta^{18}O\%$  values of karst groundwater samples are considered as the values of precipitation recharge endmembers, which are -64.56 and -9.30, respectively. The average  $\delta^2H\%$  and  $\delta^{18}O\%$  values of surface water samples are considered as the values of surface water recharge endmembers, which are -50.53 and -6.815, respectively. Using Eq. 5, the mixing ratio for all karst groundwater samples could be calculated. It should be noted that although unauthorized sewage discharge might influence groundwater isotopic values, strict pollution controls in the study area (given Baotu Spring's significance) make this factor negligible for this study.

#### 2.3. Flow monitoring and infiltration test

Based on field surveys, a large volume of released water in the Yufu River results in some flow escaping downstream and failing to infiltrate and recharge karst groundwater, thereby reducing the effective MAR rate. To verify this and investigate the infiltration capacity of MAR river segments, the flow monitoring data from multiple segments of the Yufu River and Xingji River from 2014 to 2016 were collected (Fig. 1). According to the principle of water balance, the difference in flow between the cross-sections of the MAR river segment is considered as the effective MAR rate. For Yufu River, the water is released in section #1, and the effective MAR rate equals the difference between the flow rate at section #1 and section #5 (Eq. 6). For Xingji River, the water is released in section #6, and the effective MAR rate equals the difference between the flow rate at section #6 and section #7 (Eq. 6). Then, based on the statistical data of the water release rate, the quantitative relationship between the water release rate and the effective MAR rate is analyzed.

$$\begin{cases}
Q_{Eff(Yu)} = Q_1 - Q_5 \\
Q_{Eff(Xing)} = Q_6 - Q_7
\end{cases}$$
(6)

In the equation,  $Q_{Eff(Yu)}$  and  $Q_{Eff(Xing)}$  denote the effective MAR rates of the Yufu River and the Xingji River, respectively, while  $Q_1$  through  $Q_7$  represent the flow rates of section #1 to section #7, correspondingly.

It should be noted that although the 2014~2016 flow monitoring data from two hydrological years are sufficiently representative (reflecting the stable infiltration capacity of the river channels, as no large-scale construction occurred after 2016), it remains necessary to calculate the maximum infiltration capacity to account for scenarios requiring high water release rates during extreme dry years or months. Therefore, we selected five sites along the MAR segment of Yufu River and measured the permeability coefficient of the riverbed based on in-situ double-ring infiltration test (Li et al., 2019) (Fig. 1). The infiltration test was performed at the riverbed edges (the river still maintains a small flow during the dry season). Then, the infiltration coefficient of the Yufu River MAR segments were calculated using the double-ring infiltration test results. The theoretical maximum recharge capacity was finally determined based on the river's area.

#### 2.4. Estimation of effective porosity from tracer tests

Effective porosity is a crucial parameter for simulating groundwater solute transport. The actual groundwater velocity determined by tracer tests can be used to calculate the effective porosity of karst fissured aquifers (Zuber & Motyka, 1994), using the following Equation:

$$258 n_f = \frac{\kappa_I}{\nu_t} (7)$$

This Equation is derived from Darcy's Law. In the equation, "K" represents the hydraulic conductivity (m/d), "I" is the hydraulic gradient, and " $v_i$ " is the actual groundwater velocity (referring to the advective flow velocities governing the transport). Two large-scale tracer tests were conducted at Cuima village in year 1989 and in Xingji River in year 2016 (Zhu et al., 2020). To determine the parameters required in the groundwater solute transport model, the actual groundwater flow velocity and the effective porosity of the karst aquifer were calculated based on the data from the two tracer tests (Fig.

3).

First, three groundwater flow lines were extracted from the groundwater flow field, and several calculation points for groundwater flow velocity and effective porosity were selected at equal intervals. Flow lines #1 and #2 represent the diffusion direction of the tracers from Cuima Village, while flow line #3 represents the diffusion direction of the tracers from Xingji River. Using the isochrone map of tracer peak concentration diffusion, the actual groundwater flow velocity was calculated based on the horizontal distance between adjacent isochrones at each calculation point. The hydraulic gradient at the calculation points on flow lines #1 and #2 was calculated using 1989 groundwater level monitoring data, and the hydraulic gradient on flow line #3 was calculated using 2016 data. The permeability coefficient (*K*) for each calculation point was determined using the groundwater flow model established in this research.

Fig. 3 Tracer tests conducted in Cuima Village and Xingji River

## 2.5. Groundwater flow-solute transport simulation

Numerical simulation is used to predict groundwater level and quality in this research. As mentioned in the introduction, the EPM model without inserting embedded karst-conduits is capable for groundwater flow simulation in karst regions with low development like the northern China karst areas. To verify this, we identified some typical karst fissure outcrops in the Ordovician limestone exposure area (Fig. 4) and measured the mechanical apertures of the fissures. The measurements show that the maximum mechanical aperture of the karst fissures is approximately 6 mm, while the minimum is less than 1 mm. For natural karst fissures, the hydraulic aperture used for flow calculations is typically much smaller than the mechanical aperture, with their ratio (generally less than 0.15 for karst fissures) determined by the fissure geometry and filling characteristics (Zhang & Nemcik, 2013; Zimmerman & Bodyarsson, 1996). For conservatism, we set the mechanical aperture at 6 mm and the ratio of hydraulic aperture to mechanical aperture at 0.15 to determine the maximum Reynolds number (Re). Based on the findings in Section 3.3 (Tab.1 provided in the Supplement), the maximum actual groundwater flow velocity in the study area's runoff zone is approximately 216 m/d (0.0025 m/s). Using these data, Eq. 8 yields a rough estimate indicating that the Re of karst groundwater flow in the study area ( $\leq 2.24$ ) is significantly lower than the critical Re (2000). Therefore, the flow regime in the karst fissures is laminar, justifying the use of the EPM model for simulation. Consequently, the GMS software was used to establish a karst groundwater flow and solute transport model for the Baotu Spring area. The MODFLOW 2005 and MT3DMS packages were employed to solve the groundwater flow and solute transport equations using the finite difference method.

Fig. 4 Outcrop of karst fissures in the study area

$$Re = \frac{\rho vL}{\mu} \le \frac{998 \times 0.0025 \times 0.006 \times 0.15}{1.002 \times 10^{-3}} = 2.24$$
 (8)

Where, *Re*: Reynolds number (dimensionless);

 $\rho$ : Fluid density (kg/m<sup>3</sup>);

v: Characteristic flow velocity (m/s);

L: Characteristic length (m), defined here as the hydraulic aperture of the fissures;

 $\mu$ : Fluid dynamic viscosity (kg/(m.s)).

The numerical model encompasses the Baotu Spring area, simplifying the stratigraphy into four units: Quaternary porous phreatic aquifer, intrusive rock aquitard, Ordovician-Cambrian karst aquifer, and the aquitard below the Mantou Formation (Fig. 5(a)). The Ordovician-Cambrian karst aquifer is the main aquifer and the target aquifer for MAR and groundwater extraction. In Fig. 5(a), the vertical (Z-axis) scale is exaggerated fivefold to enhance the visualization of topographic undulations and stratigraphic profile variations. The boundaries of the Ordovician-Cambrian karst aquifer are delineated in Fig. 5(b). Additionally, the boundaries of other strata are all impervious.

The model's source and sink terms include precipitation recharge, MAR from rivers and wells, groundwater extraction, spring discharge, agricultural irrigation extraction, and agricultural irrigation reinfiltration. Precipitation recharge is calculated based on precipitation quantity, infiltration recharge coefficient, and recharge zone area. The infiltration recharge coefficient accounts for surface lithology, urbanization, and agricultural development. The MAR from rivers mainly occur through the Yufu and Xingji Rivers, and the effective MAR rate is discussed in Section 3.2. The locations of MAR wells, groundwater extracting wells and MAR river segments are displayed in Fig. 5(b).

The hydraulic conductivity K consists of horizontal hydraulic conductivity  $K_x$  and vertical hydraulic conductivity  $K_y$ . The zoning and values of  $K_x$  are mainly based on the hydrogeological tests, and then identified and verified using groundwater level monitoring well data.  $K_y$  is uniformly set to 0.1 times  $K_x$ . The remaining hydrogeological parameters (specific yield, storativity, and dispersivity) are taken as empirical values. The determination of effective porosity has been discussed in Section 2.4. The simulation period spans from January 1, 2020, to December 31, 2022, with each stress period lasting one

327 month.

Next, the effect of MAR and groundwater extraction on the dynamics of karst groundwater levels was quantitatively analyzed using a groundwater flow model, with Baotu Spring's water level serving as a representative indicator. The considered MAR are from Yufu River, Xingji River, and MAR wells. For the simulation period of 2020-2022, the net variations of Baotu Spring water level caused by MAR and groundwater extraction were calculated.

Finally, to quantitatively compare the effects of various MAR and groundwater extraction on the dynamics of Baotu Spring water level, the water level net variation after 1, 2, and 3 years of continuous recharge and extraction at a constant flow rate was calculated and compared.

(a) Geological model

(b) Settings of groundwater flow model

Fig. 5 Geological model and groundwater flow model

For the solute transport simulation, sulphate is selected as the representative solute because its concentration in the surface water used for MAR is higher than in the groundwater, while the concentrations of other solute components are either similar to or lower than those in the groundwater. The simulation is based on the groundwater flow model spanning 2020 to 2022, with all recharge/discharge values averaged over this period to mitigate seasonal flow variations. The initial sulphate concentration in karst groundwater is uniformly set at 50 mg/L, reflecting the average concentration in high-quality water from the wells in western suburbs. The model then simulates the dynamics of sulphate concentration in karst groundwater after 2, 6, and 18 months of continuous recharge with sulphate concentrations of 150 mg/L, 250 mg/L, and 350 mg/L in the MAR water.

# 3. Results and Discussion

#### 3.1. Mixing percentages of groundwater recharge sources

A scatter plot of the  $\delta^2 H\%$  and  $\delta^{18} O\%$  of groundwater and surface water is generated in Fig. 6. It

shows that karst groundwater samples are distributed near the LMWL, indicating that the karst groundwater in the study area originates from precipitation (Liu et al., 2021). The isotopic enrichment of <sup>2</sup>H and <sup>18</sup>O in surface water samples is significantly higher than that in karst groundwater samples, exhibiting a typical evaporation effect. Additionally, the karst groundwater samples gradually deviate from the LMWL with the enrichment of <sup>2</sup>H and <sup>18</sup>O, indicating the mixing of precipitation and surface water. This suggests that the karst groundwater is significantly recharged by surface water.

Fig. 6 Scatter plot of  $\delta^2 H\%$  and  $\delta^{18} O\%$  in water samples

Previous studies have quantitatively calculated the contribution rates of groundwater flow from different strata to the four major springs in the Baotu Spring area of Jinan City, demonstrating varying groundwater circulation depths among these springs (Zhu et al., 2020). However, despite originating from different stratigraphic layers, the ultimate source of groundwater flow remains precipitation and surface water (Guo et al., 2019). To better evaluate MAR effects primarily conducted through river channels, it is essential to determine the proportion and spatiotemporal

distribution characteristics of surface water recharge in groundwater—an aspect not addressed in prior research.

In this study, the mixing percentage of surface water recharge in groundwater is calculated with Eq. 5 and exhibited in Fig. 7. According to the result, the closer the distance to the MAR segments of Yufu River and Xingji River is, the higher the mixing percentage of surface water recharge in groundwater. In the southern metamorphic rock and Cambrian Zhangxia Formation limestone outcrop areas, as well as the northwestern Yellow River alluvial plain, the mixing percentage of surface water recharge is generally less than 20%. In contrast, in the middle and lower segments of the Yufu River and North Dasha River basins, as well as the Xingji River basin, the mixing percentage of surface water recharge is relatively high. The highest mixing percentage is near the MAR segments of the Yufu River and Xingji River. For example, in the villages along the MAR segment of the Yufu River, the mixing percentage of surface water recharge in wells ZhaiET, Cui1, and J97 exceeds 80%, while in wells A2-30 and Ji1 near the Xingji River MAR segment, and the Springs downstream, it exceeds 50%. These results highlight that the MAR from the Yufu River and Xingji River is a significant component of karst groundwater resources, emphasizing the importance of MAR projects in ensuring groundwater resources and raising regional groundwater levels.

Fig. 7 Mixing percentage of surface water recharge in groundwater

# 3.2. Infiltration efficiency of MAR river segments

Firstly, in order to investigate the relationship between effective MAR rates and water release rates, we analyzed flow data from 2014 to 2016 (Fig. 8). Since the MAR segment of the Yufu River is divided into four segments by five flow monitoring sections, whereas Xingji River has only one MAR segment due to a single upstream and downstream section, the upstream and downstream flow rates were analyzed separately for each segment to assess groundwater infiltration capacity. Based on the data in Fig. 8, we plotted the flow relationships between upstream and downstream sections for each segment (Fig. 9, where  $Q_1 \sim Q_7$  represent the flow rates of section #1~section #7, in units of  $10^4$  m³/d).

Fig. 8 Flow rate curves at various sections of the Yufu River and Xingji River

Fig. 9 Graphical representation of the flow rate relationship between upstream and downstream

sections in MAR river segments

402

419

420

421

422

In Fig. 9(a), the single blue data point shows Q2 significantly exceeds Q1, indicating higher downstream flow than upstream flow, which suggests an additional recharge source between section #1 and section #2. Thus, this point was excluded from the fitting function. Similarly, four blue points in Fig. 9(b) that deviate markedly from the fitted line were also excluded. Furthermore, Figures 9(b)-9(e) show that when upstream flow is low, downstream flow is nearly zero, suggesting complete infiltration of river water into groundwater below a certain threshold of flow, defined as the "critical flow rate." Data analysis reveals that when flow exceeds this critical flow rate (Eq. 9), upstream and downstream flows generally follow a linear relationship (Eq. 9), with Pearson R<sup>2</sup> all exceeding 0.810.

$$\begin{cases} (1). \ Q_2 = 0.7961 \times Q_1 \ (Q_1 \ge 0, \ R^2 = 0.962) \\ (2). \ Q_3 = 0.6223 \times Q_2 - 4.424 \ (Q_2 > 7.109, \ R^2 = 0.810) \\ (3). \ Q_4 = 0.6032 \times Q_3 - 1.487 \ (Q_3 > 2.465, \ R^2 = 0.954) \\ (4). \ Q_5 = 0.8678 \times Q_4 - 1.694 \ (Q_4 > 1.952, \ R^2 = 0.952) \\ (5). \ Q_7 = 0.7863 \times Q_6 - 1.572 \ (Q_6 > 2, \ R^2 = 1) \end{cases}$$

Then, by combining Eq. 9 with Eq.6, the following relationships are established: the effective MAR 414 rate in the Yufu River is quantitatively related to  $Q_I$  (Eq. 10), while that of the Xingji River correlates 415 with  $Q_6$  (Eq. 11).

$$\begin{cases} Q_{Eff(Yu)} = Q_1 & (Q_1 \le 20.44) \\ Q_{Eff(Yu)} = 0.7449Q_1 + 5.214 & (Q_1 > 20.44) \end{cases}$$

$$\begin{cases} Q_{Eff(Xing)} = Q_6 & (Q_6 \le 2) \\ Q_{Eff(Xing)} = 0.2137Q_6 + 1.573 & (Q_6 > 2) \end{cases}$$
(11)

$$\begin{cases}
Q_{Eff(Xing)} = Q_6 & (Q_6 \le 2) \\
Q_{Eff(Xing)} = 0.2137Q_6 + 1.573 & (Q_6 > 2)
\end{cases}$$
(11)

In the equation, the units of  $Q_1$  to  $Q_7$ , as well as  $Q_{Eff(Yu)}$  and  $Q_{Eff(Xing)}$ , are all  $10^4$ m<sup>3</sup>/d.

According to Eq. 10, when the water release rate does not exceed 20.44×10<sup>4</sup>m<sup>3</sup>/d, the effective MAR rate for Yufu River equals the water release rate, indicating full infiltration of surface water before section #5. However, when the water release rate exceeds 20.44×10<sup>4</sup>m<sup>3</sup>/d, the effective MAR rate is less than the water release rate, as some surface water flows past section #5 without complete infiltration. Similarly, Eq. 11 demonstrates that the Xingji River follows a similar pattern to the Yufu River.

According to the test results of in situ double-ring infiltration test, the streambed permeability coefficient of the Yufu River MAR segment ranges from 1.96 to 2.76 m/d, with an average of 2.30 m/d across five sites. According to high-resolution satellite images, the Yufu River MAR segments (Fig. 1), from section #1 to section #5, approximately covers an area of 0.5 km². Assuming a vertical infiltration hydraulic gradient of 1, the theoretical maximum MAR rate for the Yufu River MAR segments, calculated using Darcy's Law, is approximately 114.9×10<sup>4</sup>m³/d. It should be noted that during the monitoring period (2014–2016), the maximum flow rate of the Yufu River was 34.73×10<sup>4</sup> m³/d, much less than this value. This indicates that although a water release rate exceeding 20.44×10<sup>4</sup> m³/d may lead to partial waste of recharge water, further increasing the water release rate can still enhance groundwater recharge.

Finally, the effective MAR rates of the Yufu River and Xingji River from 2020 to 2022 were calculated, as shown in Fig. 10. These calculations were used as surface water recharge inputs for the groundwater flow model.

(a) Yufu River

(b) Xingji River

Fig. 10 Water release rate VS. Effective MAR rate in Yufu River and Xingji River

## 3.3. Effective porosity of karst aquifers estimated from tracer tests

As discussed in Section 2.4, the effective porosity at each calculation point was calculated using Eq. 7, with the process and results shown in Tab.1 (provided in the Supplement). According to Tab.1, the groundwater flow velocity in the study area ranges from 52.4 to 216 m/d, and the effective porosity of the aquifer varies widely, with the maximum, minimum, and average values being 4.39×10<sup>-4</sup>, 1.28×10<sup>-5</sup>, and 1.08×10<sup>-4</sup>, respectively. Consistent with these findings, the effective porosity of Cretaceous Chalk in northeastern England's Yorkshire ranges from 3.7×10<sup>-4</sup> to 4.1×10<sup>-3</sup> (Agbotui et al., 2020), while Jurassic Limestone and Magnesian Limestone exhibit values of 1×10<sup>-4</sup> (Foley et al., 2012) and 3×10<sup>-4</sup> (Medici et al., 2019), respectively. Studies have indicated that the effective porosity of karst systems exhibits significant scale effects, primarily attributed to the heterogeneity of groundwater flow velocities caused by the non-uniform development of karst conduits and fissures. For regional groundwater studies, large-scale tracer tests and dilution tests should be employed to determine the effective porosity, which typically ranges between 10<sup>-4</sup> and 10<sup>-3</sup>—considerably lower than previously recommended values (0.1–

0.01) (Medici & West, 2021). In this paper, the average effective porosity (1.08×10<sup>-4</sup>) from all calculated points on three flow lines from the Cuima Village and Xingji River tracer tests was used to represent the karst aquifer's effective porosity for groundwater solute transport modeling.

# 3.4. Effects of MAR and extraction on groundwater level

According to the identification and verification of the groundwater flow model, the horizontal hydraulic conductivity of karst aquifers is shown in Fig. 11(a). The calculated and observed groundwater flow field as of December 31, 2022, are shown in Fig. 11(b), and the calculated and observed groundwater levels for the representative monitoring wells are shown in Fig. 11(c) to 11(f) (provided in the Supplement).

(a) Horizontal hydraulic conductivity of karst aquifers (Unit: m/d)

(b) Groundwater flow field (Unit of water level: m)

Fig. 11 Identification and verification result of the groundwater flow model

The net variations of Baotu Spring water level caused by MAR and groundwater extraction were simulated and are displayed in Fig. 12. The "net variations in the Baotu Spring water level" in Fig. 12 refers to the portion of groundwater level fluctuation in Baotu Spring caused by MAR and groundwater extraction. These variations are calculated using a numerical model based on actual MAR and groundwater extraction data. It shows that the water level of Baotu Spring rises with MAR and drops with groundwater extraction. There is also a lag in the effect of these factors on the water level.

Fig. 12 Net variations in the Baotu Spring water level caused by MAR and groundwater extraction (based on data in 2020-2022)

The water level net variation following continuous recharge and extraction at a constant flow rate over 1, 2, and 3 years was computed and compared, with results presented in Tab.2. Due to maximum recharge capacity limitations, scenarios exceeding 5×10<sup>4</sup> m³/d for Xingji River and MAR wells were excluded. Tab.2(a) demonstrates the impact of MAR on Baotu Spring's water level, revealing that well-based MAR yields the greatest effect, followed by Xingji River, while Yufu River exhibits the least influence despite its substantially higher maximum recharge capacity. Notably, well-based MAR induces minimal water level variation beyond the first year, attributable to their proximity to Baotu Spring (0.75 km, 0.81 km, and 0.35 km) and the high permeability (k=150 m/d) of the shared karst aquifer. Optimal MAR source selection should consider operational strategies: well-based MAR, despite limited capacity, is most effective for rapid short-term water level elevation; Xingji River MAR, with moderate efficacy

and capacity, suits sustained minor water level augmentation when combined with well-based MAR; whereas Yufu River MAR, though least efficient, offers substantial capacity and should serve as the primary source for significant long-term water level increases in conjunction with other sources.

Tab.2(b) delineates groundwater extraction effects on Baotu Spring's water level, indicating that eastern suburbs wells exert the strongest influence, followed by western suburbs wells, with western Jinan wells showing minimal impact. Given recorded extraction rates (2020–2022) of 5,339 m³/d (western suburbs), 70,852 m³/d (western Jinan), and 46,487 m³/d (eastern suburbs), strategic redistribution of extraction from eastern to western suburbs wells could mitigate Baotu Spring's water level decline while meeting regional groundwater demand. This approach capitalizes on the underutilized extraction potential of western suburbs wells.

Tab. 2 The effect on groundwater level net variation at Baotu Spring resulting from a constant rate of continuous MAR and groundwater extraction over 1 year, 2 years, and 3 years

(a). The effect on groundwater level rise resulting from MAR

| MAR rate ( $\times 10^4 \text{m}^3/\text{d}$ ) |    | Duration of groundwater recharge |         |         |
|------------------------------------------------|----|----------------------------------|---------|---------|
|                                                |    | 1 year                           | 2 years | 3 years |
| Yufu River                                     | 2  | 19 mm                            | 31 mm   | 36 mm   |
|                                                | 5  | 49 mm                            | 76 mm   | 90 mm   |
|                                                | 10 | 97 mm                            | 152 mm  | 179 mm  |
|                                                | 20 | 194 mm                           | 300 mm  | 353 mm  |
|                                                | 30 | 266 mm                           | 409 mm  | 480 mm  |
| Xingji River                                   | 2  | 23 mm                            | 60 mm   | 89 mm   |
|                                                | 5  | 38 mm                            | 100 mm  | 148 mm  |
| MAR wells                                      | 2  | 85 mm                            | 86 mm   | 86 mm   |
|                                                | 5  | 212 mm                           | 214 mm  | 216 mm  |

(b). The effect on groundwater level drop resulting from groundwater extraction

| Groundwater extraction rate         |    | Duration of groundwater extraction |         |         |  |
|-------------------------------------|----|------------------------------------|---------|---------|--|
| $(\times 10^4 \text{m}^3/\text{d})$ |    | 1 year                             | 2 years | 3 years |  |
| Western suburbs wells               | 5  | 115 mm                             | 138 mm  | 142 mm  |  |
|                                     | 10 | 230 mm                             | 283 mm  | 293 mm  |  |
| Western Jinan wells                 | 5  | 87 mm                              | 106 mm  | 107 mm  |  |

|                       | 10 | 175 mm | 218 mm | 221 mm |
|-----------------------|----|--------|--------|--------|
| Eastern suburbs wells | 5  | 199 mm | 322 mm | 382 mm |
|                       | 10 | 397 mm | 643 mm | 765 mm |

# 3.5. Effects of MAR on groundwater quality

Based on the groundwater solute transport model established in this paper, simulations were conducted to monitor the dynamics of sulphate concentration in karst groundwater after continuous recharge of water with sulphate concentrations of 150 mg/L, 250 mg/L, and 350 mg/L over periods of 2 months, 6 months, and 18 months. As shown in Fig. 13 (The other subfigures of Fig. 13 are provided in the Supplement), with prolonged recharge duration and deteriorating recharge water quality, the sulphate concentrations in karst groundwater increase and the affected area of karst groundwater quality expands continuously, indicating an increasing effect of MAR on karst groundwater quality over time. Therefore, deteriorating water quality from MAR poses risks to groundwater, and strict control and monitoring of recharge water quality are necessary. Additionally, sulphate concentrations in karst groundwater reach stability after 12-18 months of continuous recharge, and a linear regression established a quantitative relationship between the sulphate concentrations in karst groundwater and in MAR water (Fig. 14). In practice, target values for sulphate concentrations in karst groundwater should be preset, and the minimum control standards for sulphate concentrations in MAR water could be calculated using the relationship shown in Fig. 14. For example, according to China's Groundwater Quality Standards, the sulfate concentrations in Class I, II, and III groundwater must not exceed 50, 150 and 250 mg/L, respectively. Thus, it can be calculated that to ensure karst groundwater meets Class I, II, and III standards, the sulfate concentration in the MAR water source

must not exceed 56.5, 197.8 and 339.1 mg/L, respectively.

Fig. 13 Evolution of sulphate concentration in karst groundwater under MAR

Fig.14 Linear relationships between sulphate concentrations in karst groundwater and MAR water

#### 4. Conclusions

This study focuses on temperate semi-arid fissured karst systems, proposing an integrated multimethod quantitative approach combining isotopic analysis, flow monitoring, tracer tests, and numerical modeling. The methodology was developed to investigate the impacts of managed aquifer recharge (MAR) and extraction on karst groundwater level and quality dynamics, with a case study conducted in the Baotu Spring karst system, Jinan, China. The main conclusions are summarized as follows: (1) The conventional two-endmember mixing model for recharge estimation was enhanced by integrating  $\delta^2 H$  (‰) and  $\delta^{18}O$  (‰) data from both surface water and groundwater, which reduced uncertainties in estimating groundwater recharge ratios from multiple sources. The calculated mixing ratios of groundwater recharge indicate that surface water accounts for over 80% and 50% of groundwater recharge near the MAR segments of Yufu River and Xingji River, respectively. (2) The relationship between effective MAR rates and water release rates was quantified through flow monitoring and infiltration tests, thereby improving recharge rate quantification accuracy. Results indicate that when water release rates surpass a critical threshold (20.44×10<sup>4</sup> m<sup>3</sup>/d for Yufu River and 2×10<sup>4</sup> m<sup>3</sup>/d for Xingji River), partial surface water flows downstream, diminishing the effective MAR rate. (3) Based on large-scale regional tracer tests, the effective porosity of the investigated karst aquifer was estimated to be approximately 1.08×10<sup>-4</sup>, which enhances the reliability of solute transport simulations. This value is comparable to results reported from similar karst terrains in Europe. (4) Using image data from exposed fissures to measure apertures, a maximum Reynolds number  $(Re\approx2.24)$  for karst groundwater flow was calculated, confirming laminar flow conditions and validating the EPM model's applicability for the studied karst system. (5) Groundwater flow and solute transport

modeling was employed to assess the effects of MAR and extraction on groundwater levels and quality.

The results indicate that MAR significantly raises karst groundwater levels, though efficiency varies by 556 different MAR sources. Prolonged recharge with poor-quality MAR water may degrade groundwater 557 quality, and the maximum allowable sulfate concentrations in MAR water to meet China's Class I, II, and III groundwater standards are 56.5 mg/L, 197.8 mg/L, and 339.1 mg/L, respectively. 558 559 Overall, the methodology proposed in this study effectively analyzes the impacts of MAR and 560 extraction on groundwater level and quality. The integrated approach leverages multi-source data to 561 achieve quantitative results. These findings provide a reference for MAR implementation in temperate 562 semi-arid fissured karst systems with hydrogeological conditions similar to the Baotu Spring area. 563 **Statements and Declarations** 564 Data availability 565 The data underlying this article were provided by 801 Institute of Hydrogeology and Engineering 566 Geology, Shandong Provincial Bureau of Geology & Mineral Resources by permission. Data will be 567 shared on request to the corresponding author with permission. **Author Contribution** 568 569 The respective contributions of all authors to this paper are as follows: 570 Cao H.: Conceptualization, Methodology, Writing-Original Draft. 571 Dong W. H.: Writing-Review and Editing, Supervision, Project administration. 572 Hu C. P., Chen H. L.: Resources, Funding acquisition, Investigation, Data Curation. 573 Qian J. L.: Investigation, Data Curation. 574 Liu C. W., Lyu M. H., Gao S.: Resources, Funding acquisition.

# **Competing Interests**

The authors have no relevant financial or non-financial interests to disclose.

## Acknowledgments

- Many thanks to the sponsoring organizations (801 Institute of Hydrogeology and Engineering
- Geology, Shandong Provincial Bureau of Geology & Mineral Resources) that made this research possible.

#### Financial support

584

- This manuscript was funded by National Key Research and Development Program (No.
- 2024YFC3713100), National Natural Science Foundation of China (42202294) and Shandong Provincial
- Natural Science Foundation (ZR2021QD084).

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
