# Peer review of "Managed aquifer recharge and extraction effects on groundwater level and quality"

_EGUsphere, 2025_

## Author Response (AR1)

**CC1**:**

**General comments**

Very good modelling research in the field of karst hydrology. Please, follow my guidance to improve the manuscript.

**Specific comments**

Lines 74-76. "In karst regions with low development, such as the karst areas in northern China, groundwater flow is predominantly laminar, largely complying with Darcy's law". Insert supporting references for dominance of darcian flow in poorly karstified carbonate rocks in regions outside China:

- Agbotui P.Y., West L.J., Bottrell S.H. 2020. Characterisation of fractured carbonate aquifers using ambient borehole dilution tests. Journal of Hydrology, 589, 125191
- Medici G., Munn J.D., Parker B.L. 2024. Delineating aquitard characteristics within a Silurian dolostone

aquifer using high-density hydraulic head and fracture datasets. Hydrogeology Journal, 32(6), 1663-1691.

**Response:** The above two supporting references have been inserted in line 105-106 (unmarked manuscript):

Line 117. Specify the 3 to 4 specific objectives of your research by using numbers (e.g., i, ii, and iii).

**Response:** 4 specific objectives of this research have been specified in lines 126-133:

The specific objectives of this research are as follow: (1) To determine the sources of groundwater recharge and quantify the mixing ratios and spatial distribution of recharge using multisource data. (2) To quantify the effective infiltration recharge of the MAR segments under varying water release rates for groundwater flow modeling inputs. (3) To estimate the effective porosity of aquifers as a key parameter for groundwater solute transport modeling. (4) To establish a groundwater flow-solute transport model for the study area based on the validated the EPM model's applicability, and to quantitatively evaluate the impacts of MAR and extraction on groundwater

level and quality dynamics.

Lines 120-150. Insert information on the presence of faults that can either influence the groundwater

flow, or represent preferential pathways for the recharge.

**Response:** The information on the presence of faults has been inserted in lines 154-157:

Geologically, the study area is characterized by a northward-dipping monocline predominantly

composed of Paleozoic carbonate rock layers. Several large-scale NNW-trending faults are

developed within this area. Except for the Dongwu Fault and Mashan Fault forming the eastern and

western boundaries of the study area, respectively, the other faults are generally permeable.

Line 216. Specify that you are representing the advective flow velocities for the transport.

**Response:** It was mentioned in lines 260-261:

In the equation, "K" represents the hydraulic conductivity (m/d), "I" is the hydraulic gradient,

and " $v_t$ " is the actual groundwater velocity (referring to the advective flow velocities governing the

transport).

Lines 487-642. Please, insert the two relevant papers on darcian flow in poorly karstified and

fractured carbonates suggested above.

**Response:** The two relevant papers have been inserted in the manuscript in line 105-106.

Lines 435-465. The objectives of your modelling research appear 4 by reading your conclusions.

See my comment above in the introduction.

Response: 4 research objectives have been summarized and specified in lines 126-133, as

mentioned above.

Figures and tables

Figure 1. I can see faults in your geological cross-section. They look normal faults. But, please

explain this point in detail in study area section.

**Response:** The information on the faults have been added in lines 154-157, as mentioned above.

Figure 5a. Add the spatial scales. The vertical one cannot be detected.

**Response:** The vertical (Z-axis) scale is exaggerated fivefold to enhance the visualization of topographic undulations and stratigraphic profile variations. This has been mentioned in lines 310-312:

In Fig. 5(a), the vertical (Z-axis) scale is exaggerated fivefold to enhance the visualization of topographic undulations and stratigraphic profile variations.

Figure 6. Insert regression equations with the R2 values.

**Response:** The regression equations with the  $R^2$  values have been inserted in lines 359-360:

Fig. 6 Scatter plot of  $\delta^2 H\%$  and  $\delta^{18} O\%$  in water samples

Figure 9. Make the graphs larger and improve the graphical resolution of the figure.

**Response:** The figure has been improved in lines 436-440:

**(a) Yufu River**

Fig. 10 Water release rate VS. Effective MAR rate in Yufu River and Xingji River

Figure 11. Make letters and numbers larger.

**Response:** The figure has been improved in lines 474-481:

Fig. 12 Net variations in the Baotu Spring water level caused by MAR and groundwater extraction (based on data in 2020-2022)

**CC2:**

This paper makes a significant contribution to the field by seamlessly integrating multiple research methods—ranging from numerical simulations to isotope analyses and field tracer tests—to provide a nuanced, quantitative assessment of MAR impacts on karst groundwater systems. Its innovative approach, particularly the improved two-end-member mixing model and the rigorous evaluation of effective porosity, enhances the accuracy and reliability of the findings. Moreover, by addressing both groundwater quantity and quality in a real-world setting, the work not only advances scientific

understanding but also offers practical insights for sustainable water resource management in similar karst environments.

1. How do the authors quantify and address uncertainties in the improved two-end-member isotope mixing model, particularly given potential variations in isotopic signatures and possible contamination sources

**Response:** As you have pointed out, potential variations in isotopic signatures and possible contamination sources do indeed exist objectively. Firstly, our research group collected multiple rounds of isotopic samples from groundwater between 2021 and 2023, and the values of the two-end-member isotopes showed little variation, remaining consistent with the values mentioned in the paper. Furthermore, based on our preliminary investigations and research, we believe that the amount of water discharged from potential pollution sources is relatively small. While it might affect the nitrogen and sulfur isotopes of the groundwater, it is unlikely to have a significant impact on the hydrogen and oxygen stable isotopes. Regarding the issue you raised, an explanation has been added in lines 225-227 (unmarked manuscript):

It should be noted that although unauthorized sewage discharge might influence groundwater isotopic values, strict pollution controls in the study area (given Baotu Spring's significance) make this factor negligible for this study.

Additionally, we further elaborate on the specific differences and improvements between the improved two-end-member isotope mixing model and the traditional mixing model in lines 207-220, demonstrating that the modified model reduces uncertainty:

Due to the complexity of hydrogeological conditions (there may be unknown recharge sources affecting groundwater isotope values) and the limitations in endmembers selection (isotopic values of precipitation and surface water also vary across different regions), groundwater samples do not completely fall on the mixing line between two end-members in the  $\delta^2 H\% - \delta^{18}O\%$  diagram. For certain samples located far from the mixing line (such as Point A in Fig.2), calculating the mixing ratio using Eq. 3 or 4 essentially involves projecting sample Point A along the X- or Y-axis to Points A(3) or A(4), respectively, which may lead to significantly different results. To address this issue, this study proposes a method for computing the mixing ratio by projecting groundwater sample points onto the two-endmember mixing line in the  $\delta^2 H\% - \delta^{18}O\%$  diagram (it is reasonable to assume that using the closest point on the mixing line, i.e., the orthogonal projection of the sample point A(5),

yields a more reliable mixing ratio). The derived Equation for calculating the mixing ratio is as follows:

$$\begin{cases}
\eta_{S} = \frac{(x_{g} - x_{p})(x_{s} - x_{p}) + (y_{g} - y_{p})(y_{s} - y_{p})}{(x_{s} - x_{p})^{2} + (y_{s} - y_{p})^{2}} \\
\eta_{p} = \frac{(x_{g} - x_{g})(x_{s} - x_{p}) + (y_{s} - y_{g})(y_{s} - y_{p})}{(x_{s} - x_{p})^{2} + (y_{s} - y_{p})^{2}}
\end{cases} (5)$$

Fig.2 Schematic diagram illustrating the method for calculating two-endmember mixing proportions

- 2. The study relies on an effective porosity value derived from tracer tests; how sensitive are the solute transport simulation results to variations in effective porosity and hydraulic conductivity assumptions, and how is this uncertainty evaluated?
- 3. What are the limitations of the groundwater flow model in capturing the spatial and temporal heterogeneities of the karst system, and how robust is the model calibration and validation against observed data?

**Response** (for the above two comments): Effective porosity and hydraulic conductivity are critical factors influencing solute transport simulation results. This study characterizes the spatial heterogeneity of karst systems by assigning hydraulic conductivity to distinct zones, calibrated using observational data. An analysis of water level fitting in the numerical model is provided in the Supplement, showing that the discrepancies between observed and calculated water levels generally

remain within 1 m, with all wells successfully reproducing the overall trends of actual water level fluctuations (Figs. 11c-11f):

Fig. 11 Identification and verification result of the groundwater flow model

Figures 11(c)–11(f) present the water-level fitting results of monitoring wells in the groundwater flow numerical model of the study area. Among them, Baotu Spring (Fig. 11(c)) and Qiaozili (Fig. 11(d)) are located in the discharge zone, while M82 (Fig. 11(e)) and A2-30 (Fig. 11(f)) are situated in the recharge zone. The numerical model successfully replicates the observed hydraulic head dynamics. The maximum discrepancies between simulated and observed water levels are (1) 0.539 m for Baotu Spring, (2) 0.975 m for Qiaozili, (3) 0.883 m for M82 (after excluding abnormal fluctuations in observed data), and (4) 4.460 m for A2-30. The larger discrepancy at A2-30 is attributed to its location in a mountainous area with high and highly fluctuating groundwater levels. For all other monitoring wells, the differences remain below 1 m, indicating that the numerical model reliably simulates groundwater dynamics and is suitable for managed aquifer recharge (MAR) studies.

Furthermore, rather than partitioning effective porosity into different zones, we adopted averaged values from tracer tests as a simplified approach, based on: (1) inherent uncertainties in

hydraulic conductivity (a key input parameter for calculating effective porosity), and (2) insufficient calibration data for effective porosity.

4. The analysis of infiltration efficiency uses empirical relationships (e.g., formulas (7) and (8)) to relate released water volume to actual recharge; under what conditions might these relationships fail, and how do variable hydrological conditions affect their applicability?

**Response:** The empirical relationship between water release rate and actual recharge rate proposed in this study is derived from field flow monitoring data. Consequently, when the water release rate exceeds the monitored range, the accuracy of this empirical relationship may decrease. Nevertheless, to account for potential recharge scenarios beyond the monitoring range, we have additionally calculated the theoretical maximum infiltration capacity of the riverbed through infiltration tests, which serves as supplementary reference. This was mentioned in lines 244-249:

It should be noted that although the 2014~2016 flow monitoring data from two hydrological years are sufficiently representative (reflecting the stable infiltration capacity of the river channels, as no large-scale construction occurred after 2016), it remains necessary to calculate the maximum infiltration capacity to account for scenarios requiring high water release rates during extreme dry years or months. Therefore, we selected five sites along the MAR segment of Yufu River and measured the permeability coefficient of the riverbed based on in-situ double-ring infiltration test (Fig.1).

**And in lines 427-432:**

Assuming a vertical infiltration hydraulic gradient of 1, the theoretical maximum MAR rate for the Yufu River MAR segments, calculated using Darcy's Law, is approximately 114.9×104m3/d. It should be noted that during the monitoring period (2014–2016), the maximum flow rate of the Yufu River was 34.73×104 m3/d, much less than this value. This indicates that although a water release rate exceeding 20.44 ×104 m3/d may lead to partial waste of recharge water, further increasing the water release rate can still enhance groundwater recharge.

5. Considering the impact of MAR water quality on long-term groundwater sustainability, how do the authors define acceptable thresholds for contaminant levels, and what management strategies are proposed to mitigate the risk of groundwater quality deterioration over time?

Response: The acceptable threshold for pollutant concentrations in this study is based on China's Groundwater Quality Standards. Generally, groundwater that fails to meet the Class III water quality standard is considered unacceptable. However, in practical applications, this threshold may also be determined by the decision-makers (typically water resource management authorities) based on specific groundwater quality requirements. Additionally, this study specifies the water quality requirements for MAR to ensure that groundwater in source areas complies with Class I, II, and III

For example, according to China's Groundwater Quality Standards, the sulfate concentrations in Class I, II, and III groundwater must not exceed 50, 150 and 250 mg/L, respectively. Thus, it can be calculated that to ensure karst groundwater meets Class I, II, and III standards, the sulfate concentration in the MAR water source must not exceed 56.5, 197.8 and 339.1 mg/L, respectively.

standards under China's Groundwater Quality Standards (Lines 520-524):

6. I strongly recommend to cite below paper:

"Assimilation of sentinel-based leaf area index for modeling surface-ground water interactions in irrigation districts"

**Response:** This paper conducts in-depth and meaningful research on the numerical model of surface water-groundwater interaction. We has reference it in line 97.

**RC1:**

**General remarks**

The study described in this paper was necessary and has been done correctly.

However, the discussion and results are focused only on this case study. There is no reference of the results to the karst aquifers environment in general, so that this methodology could be adopted in other areas worldwide and the results compared. Karst areas have their own specificity, so certain principles and processes are often common in most of them. However, there are no connections in manuscript to these general features of karst aquifers.

The second shortcoming is the lack of discussion (and possibly conclusions) regarding the suitability of the adopted research methods for studies of this kind.

Response: As a typical representative area, the study region (Baotu Spring karst system in Jinan

City) of this paper exemplifies the characteristics of karst aquifer development and groundwater flow in temperate semi-arid fissured karst systems. Similar karst systems are widely distributed in Shandong, Shanxi, Shaanxi, Hebei, and Henan provinces of China. Moreover, comparable karst development patterns exist in certain regions of Europe and North America, making the research findings and methodologies applicable to these areas as well. In lines 61-85 (unmarked manuscript), we summarize the characteristics of such karst systems, supplemented with relevant literature citations and expanded discussions on this subject:

[revised manuscript text omitted]

Furthermore, we have revised the Conclusion section to summarize the integrated application and methodological improvements achieved in this study (lines 534-562):

This study focuses on temperate semi-arid fissured karst systems, proposing an integrated multi-method quantitative approach combining isotopic analysis, flow monitoring, tracer tests, and numerical modeling. The methodology was developed to investigate the impacts of managed aquifer recharge (MAR) and extraction on karst groundwater level and quality dynamics, with a case study conducted in the Baotu Spring karst system, Jinan, China.

The main conclusions are summarized as follows: (1) The conventional two-endmember mixing model for recharge estimation was enhanced by integrating  $\delta^2 H$  (‰) and  $\delta^{18}O$  (‰) data from both surface water and groundwater, which reduced uncertainties in estimating groundwater recharge ratios from multiple sources. The calculated mixing ratios of groundwater recharge indicate that surface water accounts for over 80% and 50% of groundwater recharge near the MAR segments of Yufu River and Xingji River, respectively. (2) The relationship between effective MAR rates and water release rates was quantified through flow monitoring and infiltration tests, thereby improving recharge rate quantification accuracy. Results indicate that when water release rates surpass a critical threshold (20.44×104 m³/d for Yufu River and 2×104 m³/d for Xingji River), partial surface water flows downstream, diminishing the effective MAR rate. (3) Based on large-scale regional tracer tests, the effective porosity of the investigated karst aquifer was estimated to be approximately  $1.08\times10^{-4}$ , which enhances the reliability of solute transport simulations. This value is comparable

to results reported from similar karst terrains in Europe. (4) Using image data from exposed fissures to measure apertures, a maximum Reynolds number ( $Re\approx2.24$ ) for karst groundwater flow was calculated, confirming laminar flow conditions and validating the EPM model's applicability for the studied karst system. (5) Groundwater flow and solute transport modeling was employed to assess the effects of MAR and extraction on groundwater levels and quality. The results indicate that MAR significantly raises karst groundwater levels, though efficiency varies by different MAR sources. Prolonged recharge with poor-quality MAR water may degrade groundwater quality, and the maximum allowable sulfate concentrations in MAR water to meet China's Class I, II, and III groundwater standards are 56.5 mg/L, 197.8 mg/L, and 339.1 mg/L, respectively.

Overall, the methodology proposed in this study effectively analyzes the impacts of MAR and extraction on groundwater level and quality. The integrated approach leverages multi-source data to achieve quantitative results. These findings provide a reference for MAR implementation in temperate semi-arid fissured karst systems with hydrogeological conditions similar to the Baotu Spring area.

**Detailed remarks**

Fig. 1b. - this is a hydrogeological cross-section, not "profile". It is necessary to enlarge this cross-section, because it is important for understanding the hydrogeological conditions. Precipitation is everywhere along the entire cross-section. In the zone described as "Precipitation" it is the aquifer "Recharge".

**Response:** The figure has been revised according to your suggestions in lines 145-146:

(a) Hydrogeological cross-section

Fig. 1 Geological map and hydrogeological cross-section of Baotu Spring area

Fig. 2. is unnecessary, because everything is repeated in Fig. 7 - where it is all better visible.

**Response:** Fig. 2 has been removed, with the relevant information incorporated into Fig. 5(b) in line 338-340:

Fig. 5 Geological model and groundwater flow model

Fig. 5a should be rotated by 180 degrees or 90 degrees, to be consistent with the cross-section (Fig. 1b) and to show the geological layers indicated in the lithology explanations. Now there are no visible. In addition, on Fig. 5a it will be very useful to overlay the rivers, as well as mark the MAR zone (if it is graphically possible).

**Response:** The figure has been modified to the best extent possible in accordance with your suggestions (Line 336-337).

Fig. 5 Geological model and groundwater flow model

Fig. 5b – not needed, because it repeats what is in Figures 1a and 2. Recharge coefficient values can be added (e.g. in brackets) next to the lithology explanation on Fig.1a.

**Response:** In the revised Figure 5(b), the zoning of rainfall infiltration recharge coefficients does not entirely align with that in Figure 1, as it accounts for the influence of different land use types on infiltration coefficients. Therefore, we consider this figure necessary and have integrated information from Figure 2 to streamline the content.

Fig. 6. - it is necessary to add the numbers/names of wells, boreholes, springs, etc., corresponding to the numbers in Fig. 7. If adding all of them is not possible graphically, then at least most of them.

**Response:** The figure has been revised according to your suggestions in line 359-360:

Fig. 6 Scatter plot of  $\delta^2 H\%$  and  $\delta^{18} O\%$  in water samples

Fig. 10 - maps in panels "a" and "b" are illegible, because they are too small. It is necessary to enlarge them significantly - maybe even 2x (?). No indication of units for parameters shown on these maps. Panels c, d, e, f can be moved to Supplement Material, because they do not contribute anything important enough to be in the main text.

**Response:** The figure has been revised according to your suggestions in lines 463-467.

(a) Horizontal hydraulic conductivity of karst aquifers (Unit: m/d)

(b) Groundwater flow field (Unit of water level: m)

Fig. 11 Identification and verification result of the groundwater flow model

Fig. 13 - Maps are too small. Only four of them are enough to compare the results, i.e. for 150 and 350 mg/L, for 2 and 18 months, respectively. The rest of these maps can possibly be in Supplement, if necessary.

Fig. 13 Evolution of sulphate concentration in karst groundwater under MAR

Lines 444-448 – this is an important problem, but there is no attempt to explain its cause. Either the infiltration tests were inaccurate, because they significantly overestimated the intensity of infiltration, or there is some other cause. But, there is no attempt to answer what could be the reason and what could be the way to counteract this issue.

**Response:** Regarding this point, we have reorganized the discussion on flow monitoring and infiltration tests in the Materials and Methods section (Lines 244-253) and Result and Discussion

section (Lines 427-432). First, flow monitoring establishes a quantitative relationship between water release rate and actual MAR rate. Subsequently, infiltration tests are conducted to calculate the maximum infiltration capacity, addressing scenarios requiring high water release rates during extreme dry years or months. Thus, infiltration tests serve as complementary to flow monitoring data analysis, with no contradiction between the two approaches:

It should be noted that although the 2014~2016 flow monitoring data from two hydrological years are sufficiently representative (reflecting the stable infiltration capacity of the river channels, as no large-scale construction occurred after 2016), it remains necessary to calculate the maximum infiltration capacity to account for scenarios requiring high water release rates during extreme dry years or months. Therefore, we selected five sites along the MAR segment of Yufu River and measured the permeability coefficient of the riverbed based on in-situ double-ring infiltration test (Li et al., 2019) (Fig. 1). The infiltration test was performed at the riverbed edges (the river still maintains a small flow during the dry season). Then, the infiltration coefficient of the Yufu River MAR segments were calculated using the double-ring infiltration test results. The theoretical maximum recharge capacity was finally determined based on the river's area.

Assuming a vertical infiltration hydraulic gradient of 1, the theoretical maximum MAR rate for the Yufu River MAR segments, calculated using Darcy's Law, is approximately 114.9×104m3/d. It should be noted that during the monitoring period (2014–2016), the maximum flow rate of the Yufu River was 34.73×104 m3/d, much less than this value. This indicates that although a water release rate exceeding 20.44 ×104 m3/d may lead to partial waste of recharge water, further increasing the water release rate can still enhance groundwater recharge.

Lines 458-461 – an accurate remark, but there is no determination of minimum standards for MAR water quality in this case study. It seems that this is necessary because thanks to this, this study will have an additional application effect.

**Response:** The minimum water quality criteria for MAR in this study have been established in lines 520-524, following China's Groundwater Quality Standards, as per your recommendations:

For example, according to China's Groundwater Quality Standards, the sulfate concentrations in Class I, II, and III groundwater must not exceed 50, 150 and 250 mg/L, respectively. Thus, it can be calculated that to ensure karst groundwater meets Class I, II, and III standards, the sulfate

Table 1 can be moved to the Supplement because it does not contribute anything important enough

to be in the main text.

**Response:** Table 1 has been moved to the Supplement.

RC2:

The manuscript "Managed aquifer recharge and exploitation impacts on dynamics of

groundwater level and quality in northern China karst area: Quantitative research by multimethods" presents a comprehensive analysis of groundwater dynamics in terms of head and quality

in a karst area of China. It is well-structured, well-written, and employs established methodologies.

In my opinion, the manuscript lacks novelty and generalization. I do not see it as a research article

but rather as an outstanding case study. All the methods employed in the study (i.e., isotope analysis,

infiltration tests, river flow monitoring, tracer tests, and groundwater flow and transport simulations)

have been previously introduced in the literature and extensively applied in other studies.

Specifically, I do not see any significant advancement in any of these methodologies within this

work. Moreover, the conclusions drawn from the study cannot be generalized and apply only to the

analyzed system.

That said, I want to emphasize that the work is of very high quality, and its results are indeed relevant

to the management of the Baotu Spring area. However, I believe this is not a research article in the

sense that it does not produce novel knowledge that can be generalized to other systems or develop

new methodologies.

Response: As a typical representative of a temperate semi-arid fissured karst system, the Baotu

Spring area serves as an invaluable reference for other karst regions with similar hydrogeological

conditions worldwide, including several provinces in northern China and certain regions in Europe

and North America, where similar karst aquifer media development and groundwater flow

characteristics exist, making this study potentially instructive for MAR implementation in these

areas. We have supplemented the discussion of this aspect in the Introduction to demonstrate the

generalizability of the present study (Lines 61-85 in the unmarked manuscript):

From a global perspective, significant differences exist in karst development and groundwater flow characteristics among different countries and regions. The Baotu Spring karst aquifer in Jinan, China, representing the fissured karst system in the temperate semi-arid region, exhibits remarkable hydrogeological representativeness worldwide (Liang et al., 2018). In these areas, karst aquifers typically develop in Cambrian-Ordovician carbonate formations, with their hydrogeological features being strongly controlled by geological structures. The primary aquifer medium consists of karst fissures formed by well-developed tectonic fissures, ultimately giving rise to a groundwater system dominated by an extensive network of karst fissures (Aliouache & Jourde, 2024; Jiang et al., 2022). In temperate semi-arid regions, the persistent development of dissolution is constrained by the low permeability of soluble rocks (dominated by fracture flow) and limited hydrothermal conditions, resulting in the prolonged stagnation of underground karst systems at the fracture network stage and hindering their evolution into large-scale cave or conduit systems. Moreover, such regions often feature large karst springs as concentrated discharge points of groundwater (Criss, 2010). Due to seasonal recharge fluctuations (primarily from precipitation) (Bhering et al., 2021), these springs exhibit significant discharge variations. Therefore, scientifically adjusting recharge strategies based on precipitation variability to maintain spring flow constitutes a key research issue.

In China, temperate semi-arid fissured karst groundwater systems similar to the Baotu Spring are predominantly distributed across several northern provinces, including Shandong (Liu et al., 2021), Shanxi (Zhang et al., 2018), Hebei (M. Gao et al., 2023), Henan (Yin et al., 2023) and Shaanxi (Li et al., 2020). Globally, systems exhibiting varying degrees of similarity can be observed in certain regions, notably in the U.K (Agbotui et al., 2020), France (Ballesteros et al., 2020), Germany (Knöll & Scheytt, 2017), Italy (Pagnozzi et al., 2020), the U.S (Criss, 2010) and Canada (Perrin et al., 2011). These regions all face similar challenges related to seasonal drought and karst groundwater pollution. The research on karst groundwater at Baotu Spring and its artificial recharge practices can provide valuable insights for these areas.

Furthermore, despite the abundance of existing research data and the well-established, widely applied methodologies employed in this study, previous investigations have often lacked comprehensive integration and analysis of these mature techniques and extensive datasets. This study systematically consolidates these methods and introduces targeted improvements tailored to the characteristics of temperate semi-arid fissured karst systems. The revised Introduction further

discusses the applicability and limitations of the methodologies adopted in this research (Lines 86-118):

Existing research on the effects of managed aquifer recharge (MAR) and extraction on groundwater level and quality has established relatively mature methodologies (Ringleb et al., 2016). However, most approaches remain qualitative or semi-quantitative. Hydrogeochemical and isotopic techniques are widely employed in MAR studies (Akurugu et al., 2022; Li et al., 2023). A key limitation of hydrogeochemical analysis is its susceptibility to ambiguity, as unidentified contamination sources may obscure results. Isotopic tracers are frequently used to identify recharge sources, and the integration of multiple hydrochemical and isotopic indicators (Guo et al., 2019) allows estimation of source contributions (Deng et al., 2022). Nevertheless, this method faces challenges, including uncertainty in determining precise isotopic signatures for each recharge source, which may compromise accuracy. Additionally, the scarcity of long-term isotopic monitoring data restricts the applicability of this approach for analyzing temporal variations in MAR effects.

Numerical simulation serves as an effective method for MAR quantitative analysis (Medici et al., 2021; Ostad-Ali-Askari & Shayannejad, 2021; Zafarmomen et al., 2024). The selection of simulation programs depends on karst aquifer characteristics. While conduit flow process (CFP) models are suitable for well-developed karst systems (Chang et al., 2015), their application is constrained by the requirement for detailed conduit dimension data, particularly in regional-scale modeling (Jourde & Wang, 2023). Previous studies have demonstrated the feasibility of employing a simplified equivalent porous medium (EPM) model without embedded karst conduits for regional groundwater numerical simulations in temperate semi-arid fissured karst systems with limited karst development (Kang et al., 2011; Luo et al., 2020; Scanlon et al., 2003). However, these studies often lack field investigations to verify whether groundwater flow regimes satisfy the laminar flow assumption inherent to EPM models (Agbotui et al., 2020; Medici et al., 2024).

Studies indicate that accurate estimation of effective porosity in karst aquifers is critical when simulating solute transport using the equivalent porous medium (EPM) model (Kidmose et al., 2023; Ren et al., 2018). Overestimation of effective porosity often leads to underestimated groundwater flow velocities, introducing significant errors in pollution control strategies (Medici & West, 2021; Medici et al., 2019). To improve EPM model reliability, effective porosity should be derived from regional-scale hydraulic tests (e.g., tracer test) (Medici & West, 2021; Worthington et al., 2019; Zhu

et al., 2020).

Similarly, in MAR studies using numerical simulations, precise determination of groundwater recharge rates is essential for result accuracy (Hartmann et al., 2015). For MAR driven by riverbed infiltration, methods such as infiltration tests (Xi et al., 2015) and riverflow monitoring can quantify recharge rates via hydrodynamics and water-balance principles. Cross-validation of these methods in field studies reduces uncertainty from data limitations, enhancing MAR-related quantitative assessments (Mudarra et al., 2019).

In addition, supplementary discussions have been incorporated into the Results and Discussion section to thoroughly elucidate the significance and innovations of this study:

- (1) Lines 361-368: Previous studies have quantitatively calculated the contribution rates of groundwater flow from different strata to the four major springs in the Baotu Spring area of Jinan City, demonstrating varying groundwater circulation depths among these springs (Zhu et al., 2020). However, despite originating from different stratigraphic layers, the ultimate source of groundwater flow remains precipitation and surface water (Guo et al., 2019). To better evaluate MAR effects primarily conducted through river channels, it is essential to determine the proportion and spatiotemporal distribution characteristics of surface water recharge in groundwater—an aspect not addressed in prior research.
- (2) Lines 446-454: Consistent with these findings, the effective porosity of Cretaceous Chalk in northeastern England's Yorkshire ranges from 3.7×10-4 to 4.1×10-3 (Agbotui et al., 2020), while Jurassic Limestone and Magnesian Limestone exhibit values of 1×10-4 (Foley et al., 2012) and 3×10-4 (Medici et al., 2019), respectively. Studies have indicated that the effective porosity of karst systems exhibits significant scale effects, primarily attributed to the heterogeneity of groundwater flow velocities caused by the non-uniform development of karst conduits and fissures. For regional groundwater studies, large-scale tracer tests and dilution tests should be employed to determine the effective porosity, which typically ranges between 10-4 and 10-3—considerably lower than previously recommended values (0.1–0.01) (Medici & West, 2021).

Finally, we have revised the Conclusion section to summarize the integrated application and methodological improvements achieved in this study (lines 534-562):

This study focuses on temperate semi-arid fissured karst systems, proposing an integrated multi-method quantitative approach combining isotopic analysis, flow monitoring, tracer tests, and

numerical modeling. The methodology was developed to investigate the impacts of managed aquifer recharge (MAR) and extraction on karst groundwater level and quality dynamics, with a case study conducted in the Baotu Spring karst system, Jinan, China.

The main conclusions are summarized as follows: (1) The conventional two-endmember mixing model for recharge estimation was enhanced by integrating  $\delta^2 H$  (%) and  $\delta^{18}O$  (%) data from both surface water and groundwater, which reduced uncertainties in estimating groundwater recharge ratios from multiple sources. The calculated mixing ratios of groundwater recharge indicate that surface water accounts for over 80% and 50% of groundwater recharge near the MAR segments of Yufu River and Xingji River, respectively. (2) The relationship between effective MAR rates and water release rates was quantified through flow monitoring and infiltration tests, thereby improving recharge rate quantification accuracy. Results indicate that when water release rates surpass a critical threshold (20.44×104 m3/d for Yufu River and 2×104 m3/d for Xingji River), partial surface water flows downstream, diminishing the effective MAR rate. (3) Based on large-scale regional tracer tests, the effective porosity of the investigated karst aquifer was estimated to be approximately 1.08×10-4, which enhances the reliability of solute transport simulations. This value is comparable to results reported from similar karst terrains in Europe. (4) Using image data from exposed fissures to measure apertures, a maximum Reynolds number ( $Re \approx 2.24$ ) for karst groundwater flow was calculated, confirming laminar flow conditions and validating the EPM model's applicability for the studied karst system. (5) Groundwater flow and solute transport modeling was employed to assess the effects of MAR and extraction on groundwater levels and quality. The results indicate that MAR significantly raises karst groundwater levels, though efficiency varies by different MAR sources. Prolonged recharge with poor-quality MAR water may degrade groundwater quality, and the maximum allowable sulfate concentrations in MAR water to meet China's Class I, II, and III groundwater standards are 56.5 mg/L, 197.8 mg/L, and 339.1 mg/L, respectively.

Overall, the methodology proposed in this study effectively analyzes the impacts of MAR and extraction on groundwater level and quality. The integrated approach leverages multi-source data to achieve quantitative results. These findings provide a reference for MAR implementation in temperate semi-arid fissured karst systems with hydrogeological conditions similar to the Baotu Spring area.

Some elements that should be considered and could potentially improve the quality of the work include:

1. Key studies that I see as relevant to this research are not cited, such as Mudarra & Hartmann (2019) and Hartmann et al. (2015).

**Response:** The above two papers have been inserted in line 118 and line 115.

2. Some references mentioned in the article are missing or not properly cited in the reference list, making it difficult to review the manuscript (e.g., Chuanlei Li et al., 2022, and J. Li et al., 2023).

**Response:** We have carefully re-examined and removed the missing references.

3. Be more precise when stating that simulating karst systems using the Equivalent Porous Medium (EPM) assumption is valid in karst. As written, it suggests that this approach is applicable to any non-developed system. However, some of the references you cite indicate that this assumption is valid only in specific scenarios and for particular applications (e.g., Scanlon et al., 2003). Additionally, it is difficult to support the claim that flow within karst systems is "predominantly laminar," as these systems typically contain sinkholes and caves where turbulence surely occurs. Response: Regarding the applicability of the EPM model in the study area, we have provided a detailed supplementary discussion in the Introduction section. This discussion primarily summarizes the characteristics of the temperate semi-arid fissured karst system represented by our study area, as well as case studies employing the EPM model for similar research in other global regions. Furthermore, in full consideration of your suggestions, we have conducted additional investigations on the developmental characteristics of karst fissures in the study area. Specifically, we photographed and measured the mechanical apertures of these fissures. Additionally, we roughly calculated the Reynolds number of groundwater flow based on actual flow velocities, thereby confirming that the groundwater flow in the study area exhibits laminar behavior, which satisfies the assumptions of Darcy's Law (Lines 282-306):

To verify this, we identified some typical karst fissure outcrops in the Ordovician limestone exposure area (Fig. 4) and measured the mechanical apertures of the fissures. The measurements show that the maximum mechanical aperture of the karst fissures is approximately 6 mm, while the

minimum is less than 1 mm. For natural karst fissures, the hydraulic aperture used for flow calculations is typically much smaller than the mechanical aperture, with their ratio (generally less than 0.15 for karst fissures) determined by the fissure geometry and filling characteristics (Zhang & Nemcik, 2013; Zimmerman & Bodvarsson, 1996). For conservatism, we set the mechanical aperture at 6 mm and the ratio of hydraulic aperture to mechanical aperture at 0.15 to determine the maximum Reynolds number (Re). Based on the findings in Section 3.3 (Tab. 1 provided in the Supplement), the maximum actual groundwater flow velocity in the study area's runoff zone is approximately 216 m/d (0.0025 m/s). Using these data, Eq.8 yields a rough estimate indicating that the Re of karst groundwater flow in the study area ( $\leq$ 2.24) is significantly lower than the critical Re (2000). Therefore, the flow regime in the karst fissures is laminar, justifying the use of the EPM model for simulation.

Fig. 4 Outcrop of karst fissures in the study area

$$Re = \frac{\rho vL}{\mu} \le \frac{998 \times 0.0025 \times 0.006 \times 0.15}{1.002 \times 10^{-3}} = 2.24$$
 (8)

Where, Re: Reynolds number (dimensionless);

 $\rho$ : Fluid density (kg/m3);

v: Characteristic flow velocity (m/s);

L: Characteristic length (m), defined here as the hydraulic aperture of the fissures;

 $\mu$ : Fluid dynamic viscosity (kg/(m.s)).

**4. Typo in figure 1 (Reservoirs)**

**Response:** A thorough review has been conducted to identify and rectify spelling errors in both the textual content and graphical representations.

5. In Section 2.2., it is unclear why the ratios calculated via Eqs. (3) and (4) are not appropriate for your analysis, how Eq. (5) was derived, and to what extent Eq. (5) improves upon Eqs. (3) and (4). **Response:** We have provided additional elaboration on both the interpretation of these equations and the theoretical justification for the enhanced appropriateness of the modified equations (Lines 207-220):

Due to the complexity of hydrogeological conditions (there may be unknown recharge sources

affecting groundwater isotope values) and the limitations in endmembers selection (isotopic values of precipitation and surface water also vary across different regions), groundwater samples do not completely fall on the mixing line between two end-members in the  $\delta^2 H\%-\delta^{18}O\%$  diagram. For certain samples located far from the mixing line (such as Point A in Fig. 2), calculating the mixing ratio using Eq. 3 or 4 essentially involves projecting sample Point A along the X- or Y-axis to Points A(3) or A(4), respectively, which may lead to significantly different results. To address this issue, this study proposes a method for computing the mixing ratio by projecting groundwater sample points onto the two-endmember mixing line in the  $\delta^2 H\%-\delta^{18}O\%$  diagram (it is reasonable to assume that using the closest point on the mixing line, i.e., the orthogonal projection of the sample point A(5), yields a more reliable mixing ratio). The derived Equation for calculating the mixing ratio is as follows:

Fig. 2 Schematic diagram illustrating the method for calculating two-endmember mixing proportions

6. The preprocessing steps for the streamflow data are not clearly explained. Additionally, it is unclear whether these data are representative of other years for these rivers. More details on the approach used in the analysis should be included.

**Response:** We have supplemented several figures and equations, and provided a more detailed discussion on the analysis process of river monitoring data (lines 386-423):

Firstly, in order to investigate the relationship between effective MAR rates and water release rates, we analyzed flow data from 2014 to 2016 (Fig. 8). Since the MAR segment of the Yufu River is divided into four segments by five flow monitoring sections, whereas Xingji River has only one MAR segment due to a single upstream and downstream section, the upstream and downstream flow rates were analyzed separately for each segment to assess groundwater infiltration capacity. Based on the data in Fig. 8, we plotted the flow relationships between upstream and downstream sections for each segment (Fig. 9, where  $Q_1 \sim Q_7$  represent the flow rates of section #1~section #7, in units of  $10^4$  m³/d).

Fig. 8 Flow rate curves at various sections of the Yufu River and Xingji River

Fig. 9 Graphical representation of the flow rate relationship between upstream and downstream sections in MAR river segments

In Fig. 9(a), the single blue data point shows Q2 significantly exceeds Q1, indicating higher downstream flow than upstream flow, which suggests an additional recharge source between section #1 and section #2. Thus, this point was excluded from the fitting function. Similarly, four blue points in Fig. 9(b) that deviate markedly from the fitted line were also excluded. Furthermore, Figures 9(b)-9(e) show that when upstream flow is low, downstream flow is nearly zero, suggesting complete infiltration of river water into groundwater below a certain threshold of flow, defined as the "critical flow rate." Data analysis reveals that when flow exceeds this critical flow rate (Eq. 9), upstream and downstream flows generally follow a linear relationship (Eq. 9), with Pearson R2 all exceeding 0.810.

$$\begin{cases} (1). \ Q_2 = 0.7961 \times Q_1 \ (Q_1 \ge 0, \ R^2 = 0.962) \\ (2). \ Q_3 = 0.6223 \times Q_2 - 4.424 \ (Q_2 > 7.109, \ R^2 = 0.810) \\ (3). \ Q_4 = 0.6032 \times Q_3 - 1.487 \ (Q_3 > 2.465, \ R^2 = 0.954) \\ (4). \ Q_5 = 0.8678 \times Q_4 - 1.694 \ (Q_4 > 1.952, \ R^2 = 0.952) \\ (5). \ Q_7 = 0.7863 \times Q_6 - 1.572 \ (Q_6 > 2, \ R^2 = 1) \end{cases}$$

Then, by combining Eq. 9 with Eq.6, the following relationships are established: the effective

MAR rate in the Yufu River is quantitatively related to  $O_l$  (Eq. 10), while that of the Xingji River correlates with  $Q_6$  (Eq. 11).

$$\begin{cases} Q_{Eff(Yu)} = Q_1 & (Q_1 \le 20.44) \\ Q_{Eff(Yu)} = 0.7449Q_1 + 5.214 & (Q_1 > 20.44) \end{cases}$$

$$\begin{cases} Q_{Eff(Xing)} = Q_6 & (Q_6 \le 2) \\ Q_{Eff(Xing)} = 0.2137Q_6 + 1.573 & (Q_6 > 2) \end{cases}$$
(11)

$$\begin{cases} Q_{Eff(Xing)} = Q_6 & (Q_6 \le 2) \\ Q_{Eff(Xing)} = 0.2137Q_6 + 1.573 & (Q_6 > 2) \end{cases}$$
 (11)

In the equation, the units of  $Q_1$  to  $Q_7$ , as well as  $Q_{Eff(Yu)}$  and  $Q_{Eff(Xing)}$ , are all  $10^4$ m3/d.

According to Eq. 10, when the water release rate does not exceed 20.44×104m3/d, the effective MAR rate for Yufu River equals the water release rate, indicating full infiltration of surface water before section #5. However, when the water release rate exceeds 20.44×104m3/d, the effective MAR rate is less than the water release rate, as some surface water flows past section #5 without complete infiltration. Similarly, Eq. 11 demonstrates that the Xingji River follows a similar pattern to the Yufu River.

Additionally, the primary objective of analyzing this dataset is to investigate the seepage capacity of the riverbed. Given that the seepage capacity remains consistent across other years (as no large-scale construction has been conducted on the riverbed in recent years), the representativeness of the data can be assured (Lines 244-247):

It should be noted that although the 2014~2016 flow monitoring data from two hydrological years are sufficiently representative (reflecting the stable infiltration capacity of the river channels, as no large-scale construction occurred after 2016), it remains necessary to calculate the maximum infiltration capacity to account for scenarios requiring high water release rates during extreme dry years or months.

7. How did you construct Figure 7? Is this any sort of kriging?

Response: The figure was generated using the natural neighborhood method, with appropriate extrapolation and smoothing applied to the initial interpolation curve.

8. How did you carry out the infiltration experiments? Did you perform the experiment in the riverbed or in areas close to the riverbed?

**Response:** The infiltration experiment was conducted on the riverbed, specifically along its margins (although during the dry season, a minimal water flow persisted in the central channel) (Lines 249The infiltration test was performed at the riverbed edges (the river still maintains a small flow during the dry season).

9. Why the Figure 8b indicates that in almost the entire period the infiltrated water coincides with the released water and in figure 9b the actual recharge differs from the released water volume? **Response:** During the flow monitoring period of the Xingji River (2014), the flow rate was predominantly below 20,000 m³/d, allowing complete riverine infiltration. Consequently, the water release rate equaled the actual MAR rate, except in the final month when the discharge exceeded 20,000 m³/d (resulting in a water release rate surpassing the actual MAR rate). However, during the simulation period (2020–2022), the water release rate of the Xingji River was substantially higher, frequently exceeding 20,000 m³/d. Thus, the water release rate consistently exceeded the actual MAR rate during this phase.

**References**

[revised manuscript text omitted]